# Helminth infection modulates systemic pro-inflammatory cytokines and chemokines implicated in type 2 diabetes mellitus pathogenesis

**Anuradha Rajamanickam**[1], **Saravanan Munisankar**[1], **Chandrakumar Dolla**[2], **Pradeep A. Menon**[2], **Kannan Thiruvengadam**[2], **Thomas B. Nutman**[3], **Subash Babu**[1,3,4]*

**1** National Institute of Health-NIRT-International Center for Excellence in Research, Chennai, India, **2** National Institute for Research in Tuberculosis, Chennai, India, **3** Laboratory of Parasitic Diseases, National Institute of Allergy and Infectious Diseases, National Institutes of Health, Bethesda, Maryland, United States of America, **4** Frederick National Laboratory for Cancer Research sponsored by the National Cancer Institute, Frederick, Maryland, United States of America

* sbabu@mail.nih.gov, sbabu@nirt.res.in

**Data Availability Statement:** All relevant data are within the manuscript and its Supporting Information files.

## Abstract

### Background

The prevalence of helminth infections exhibits an inverse association with the prevalence of Type 2 diabetes mellitus (T2DM), and helminths are postulated to mediate a protective effect against T2DM. However, the biological mechanism behind this effect is not known.

### Aims/Methods

We postulated that helminth infections act by modulating the pro-inflammatory cytokine and chemokine milieu that is characteristic of T2DM. To examine the association of cytokines and chemokines in helminth-diabetes co-morbidity, we measured the plasma levels of a panel of pro-inflammatory cytokines and chemokines in individuals with *Strongyloides stercoralis* infection (*Ss*+) and T2DM at the time of *Ss* diagnosis and then 6 months after definitive anthelmintic treatment along with uninfected control individuals with T2DM alone (*Ss*-).

### Principal findings

*Ss*+ individuals exhibited significantly diminished levels of the pro-inflammatory cytokines–IL-1α, IL-1β, IL-6, IL-12, IL-18, IL-23, IL-27, G-CSF and GM-CSF and chemokines–CCL1, CCL2, CCL3, CCL11, CXCL1, CXCL2, CXCL8, CXCL9, CXCL10 and CXCL11. In contrast, *Ss*+ individuals exhibited significantly elevated levels of IL-1Ra. Anthelmintic treatment resulted in increased levels of all of the cytokines and chemokines.

### Conclusions

Thus, helminth infections alleviate and anthelmintic therapy partially restores the plasma cytokine and chemokine levels in helminth-diabetes co-morbidity. Our data therefore offer a plausible biological mechanism for the protective effect of helminth infections against T2DM.

**Funding:** This research was supported [in part] by the National Institutes of Health, funded by NCI Contract No. 75N91019D00024, Task Order No. 75N91019F00131. The funders had no role in study design, data collection and analysis, decision to publish, or preparation of the manuscript.

**Competing interests:** The authors have declared that no competing interests exist.

## Author summary

Helminth infections are postulated to provide a degree of protection against the development of metabolic disorders such as T2DM and alleviate pathology following development of such disorders. However, the biological mechanism underlying this interaction is largely unknown. Since pro-inflammatory cytokines and chemokines are major drivers of pathology in T2DM, we examined the influence of coexistent helminth infection (in this case, *Strongyloides stercoralis*) on the cytokine and chemokine milieu in T2DM. We demonstrate that helminth infection significantly alleviates the pro-inflammatory milieu in T2DM by lowering the systemic levels of cytokines and chemokines. We also demonstrate that anthelmintic therapy exacerbates this pro-inflammatory milieu by partially restoring the high levels of cytokines and chemokines. So, our data uncovers a role for cytokines and chemokines in the mainstream interaction between helminth infections and metabolic disorders.

## Introduction

Helminth infections affect about one-quarter of the globe and are highly prevalent in lower to middle-income countries [1]. In contrast, the prevalence of inflammatory metabolic diseases, such as Type 2 diabetes mellitus (T2DM) is high in high-income countries. The absence of exposure to helminth infections has been postulated as one mechanism to explain this markedly increased prevalence of T2DM [2–6]. Recent observational studies in India, Indonesia, China and Australia have reported that the prevalence of helminth infections was significantly lower in T2DM individuals compared to non-diabetic controls [4, 5, 7, 8], thus confirming a protective effect of helminths against T2DM pathogenesis.

Among various pathophysiological mechanisms underlying the development of T2DM, a major one is the development of chronic, low-level inflammation, also named meta-inflammation, which is characterized by a distinctly exaggerated pro-inflammatory cytokine and chemokine milieu [9]. This ultimately leads to increased insulin resistance and perturbed glucose/lipid metabolism [10]. In contrast, helminth infections are typically characterized by the induction of Type 2 immune responses, with elevations in Type 2 and regulatory cytokines [11]. Moreover, the regulatory networks induced by helminth parasites can modulate bystander immune responses, including those of allergy and auto-immunity [12].

Therefore, we postulated that one possible mechanism by which helminth infections mediate protection against T2DM is by modulating the pro-inflammatory milieu in T2DM. To test this hypothesis, we examined a panel of pro-inflammatory cytokines and chemokines before and after anthelmintic therapy in a cohort of *Strongyloides stercoralis*–diabetes individuals and compared them to diabetic individuals alone. Our data reveal that helminth infections do indeed diminish the plasma levels of cytokines and chemokines in T2DM and this is partially reversed following chemotherapy.

## Material and methods

### Ethics statement

All the study participants were assessed as part of a natural history study protocol (12-I-073) approved by Institutional Review Boards of the National Institute of Allergy and Infectious Diseases (USA) and the National Institute for Research in Tuberculosis (India), and informed written consent was obtained from all participants. This was the same study population that was previously used for assessment of metabolic parameters [13].

## Study population

We enrolled 118 individuals comprising of 60 clinically asymptomatic *Ss*-infected individuals with T2DM (hereafter *Ss*+), and 58 individuals with T2DM and no *Ss* infection (hereafter *Ss*-) in Kanchipuram District, Tamil Nadu, South India (Table 1). These study participants were all recruited from a rural population by screening of individuals for helminth infection by stool microscopy and serology as described earlier [14–17]. All the recruited study participants were aged from 18 to 75 years. Individuals with any previous history of helminth infection or previous anthelmintic treatment or HIV infections, individuals who had iron deficiency anemia, alcoholism, chronic renal failure, hyperbilirubinaemia and those taking large doses of aspirin were excluded from the study. Pregnant or lactating women were excluded from the study.

## Parasitological examination and anthelmintic treatment

*Ss* infection was diagnosed by the presence of IgG antibodies to the recombinant NIE antigen as described earlier [15, 17]. A single stool sample was obtained and examined for intestinal helminth infection by Kato-Katz technique. Stool samples were found to be negative for other intestinal helminths by stool microscopy. Subsequently, *Ss* infection was further confirmed by specialized stool examination with nutrient agar plate cultures [18]. Filarial infection was excluded in all study participants by virtue of being negative in tests for circulating filarial antigen. All *Ss*+ study participants were treated with a single dose of ivermectin (12mg) and albendazole (400 mg) and follow–up blood draws were collected six months later. Following anthelmintic treatment, parasitological examinations were repeated after 6 months to confirm successful chemotherapy.

## Determination of T2DM status

Diabetes was defined as an HbA1c reading of 6.5% or greater and a random blood glucose of >200 mg/dl, according to the American Diabetes Association criteria. All the biochemical parameters were measured after overnight fasting except random blood glucose. All of the diabetic individuals in this study were newly diagnosed and were not on any anti-diabetic medication previously. All diabetic individuals were referred to the primary health care centre for diabetic treatment.

## Measurement of biochemical and anthropometric parameters

Anthropometric measurements, including height, weight and waist circumference, and biochemical parameters, including plasma glucose, lipid profiles and HbA1c were obtained using

**Table 1. Demographic and Biochemical parameters.**

|  | *Ss*+ | *Ss*- |
|---|---|---|
|  | **n = 60** | **n = 58** |
| M/F | 30/30 | 30/28 |
| Age | 46 (24–63) | 45 (22–63) |
| RBG (mg/dl) | 179 (140–438) | 180.5 (140–198) |
| HbA1c (%) | 8.57 (6.5–12.5) | 8.9 (6.5–11.8) |
| Urea (mg/dl) | 19.5 (12.34) | 21.9 (11–42) |
| Creatinine (mg/dl) | 0.78 (0.3–1) | 0.85 (0.6–1.0) |
| ALT (U/L) | 17.7 (7–60) | 22.4 (7–92) |
| AST (U/L) | 27.8 (16–110) | 24.7 (11–68) |

standardized techniques as described previously [19]. Serum samples were used for biochemical parameters and plasma samples were used for the other measurements.

## Measurement of plasma adipocytokines and cytokine levels

Plasma levels of cytokines: IL1α, IL-1β, IL-6, IL-12, IL-18, IL-23, IL-27, G-CSF and GM-CSF were measured using a Bioplex multiplex assay system (Bio-Rad, Hercules, CA). The levels of chemokines: CCL1, CCL2, CCL3, CCL4, CCL11, CXCL1, CXCL2, CXCL9, CXCL10 and CXCL11 were measured using Human magnetic Luminex Assay Kit from R&D Systems according to the manufacturer's protocol. Plasma level of IL-1Ra was measured using the Quantikine ELISA kit (R&D Systems) according to the manufacturer's instructions. We measured the *Ss+* and *Ss-* samples side by side for each cytokine.

## Statistical analysis

Data analyses were performed using Graph-Pad PRISM Version 8.0 (GraphPad, San Diego, CA) and JMP14 software was used to plot Principle Component Analysis (PCA) and heatmap. Central tendency was measured using Geometric means (GM). Statistically significant differences were analyzed using Mann-Whitney U tests were used to compare *Ss+* vs. *Ss-* and the Wilcoxon signed rank test was used to compare parameters before and after treatment followed by Holm's correction for multiple comparisons. The logistic regression was not suitable for this data to predict the group differences (i.e. *Ss+* and *Ss-*) because of the multicollinearity between most of the key variables. Those variables significantly different by Mann-Whitney test were also significantly different by the univariate logistics regression model. Sample size calculation was done to detect a significant difference (p<0.05) among the cytokines and chemokines based on preliminary analysis between the two groups. We determined that we needed 58 individuals in each group to detect this difference with a power of 90% and a Type I error of 5%.

## Results

### Study population characteristics

The baseline demographic characteristics and biochemical parameters of *Ss+* and *Ss-* individuals are shown in Table 1. As shown and as described previously [13], there were no significant differences in age, sex, BMI or other biochemical parameters between the two groups.

### Diminished plasma levels of pro-inflammatory cytokines in *Ss+* individuals with T2DM

To determine the effect of *Ss* infection on the pro-inflammatory cytokine milieu in T2DM, we measured the plasma levels of IL-1α, IL-1β, IL-1Ra, IL-6, IL-12, IL-18, IL-23, IL-27, G-CSF and GM-CSF in *Ss+* and *Ss-* individuals. As shown in Fig 1, *Ss+* individuals had significantly lower levels of IL-1α (GM of 350.1 pg/ml in *Ss+* vs. 458.8 pg/ml in *Ss-*; p = 0.0009), IL-1β (GM of 254.1 pg/ml in *Ss+* vs. 342.4 pg/ml in *Ss-*; p = 0.0008), IL-6 (GM of 65.58 pg/ml in *Ss+* vs. 133.3 pg/ml in *Ss-*; p = 0.0007), IL-12 (GM of 84.46 pg/ml in *Ss+* vs. 88.6 pg/ml in *Ss-*; p = 0.0042), IL-18 (GM of 283.7 pg/ml in *Ss+* vs. 412.1 pg/ml in *Ss-*; p = 0.0260), IL-23 (GM of 244.1 pg/ml in *Ss+* vs. 298.2 pg/ml in *Ss-*; p = 0.0380), IL-27 (GM of 323.8 pg/ml in *Ss+* vs. 516.3 pg/ml in *Ss-*; p = 0.0483), G-CSF (GM of 68.46 pg/ml in *Ss+* vs. 114.8 pg/ml in *Ss-*; p = 0.0426), and GM-CSF (GM of 78.89 pg/ml in *Ss+* vs. 95.22 pg/ml in *Ss-*; p = 0.0448) in comparison with *Ss-* individuals. In contrast, *Ss+* individuals had significantly higher levels of IL-1Ra (GM of 267.6 pg/ml in *Ss+* vs. 203.4 pg/ml in *Ss-*; p = 0.0004). Thus, *Ss* infection appears to modulate the systemic pro-inflammatory cytokine milieu in T2DM.

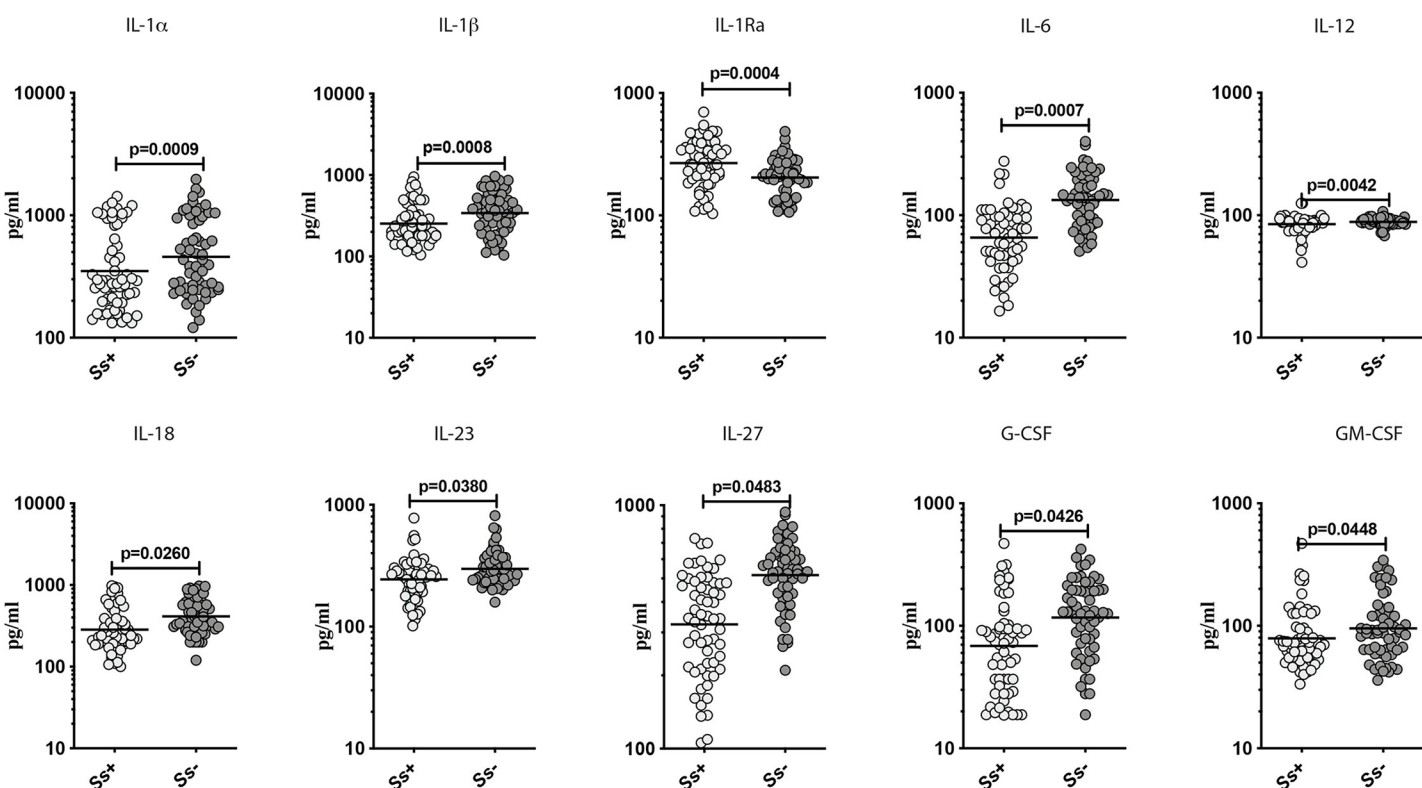

**Fig 1. Diminished plasma levels of pro-inflammatory cytokines in *Ss*+ individuals with T2DM.** Plasma levels of IL1-α, IL1-β, IL-1Ra, IL-6, IL-12, IL-18, IL-23, IL-27, G-CSF and GM-CSF cytokines were measured in *Ss*+ and *Ss*- individuals. Each dot is an individual subject with the bar representing the geometric mean (GM). p values were calculated using the Mann-Whitney U tests followed by Holm's correction for multiple comparisons.

## Diminished plasma levels of pro-inflammatory chemokines in *Ss*+ individuals with T2DM

To determine the effect of *Ss* infection on the pro-inflammatory chemokine milieu in T2DM, we measured the plasma levels of CCL1, CCL2, CCL3, CCL11, CXCL1, CXCL2, CXCL9, CXCL10 and CXCL11 in *Ss*+ and *Ss*- individuals. As shown in Fig 2, *Ss*+ individuals had significantly lower levels of CCL1 (GM of 878.3 pg/ml in *Ss*+ vs. 1095 pg/ml in *Ss*-; p = 0.0010), CCL2 (GM of 227.4 pg/ml in *Ss*+ vs. 277.9 pg/ml in *Ss*-; p = 0.0045), CCL3 (GM of 223.6 pg/ml in *Ss*+ vs. 300.9 pg/ml in *Ss*-; p = 0.0012), CCL11 (GM of 406.9 pg/ml in *Ss*+ vs. 521.2 pg/ml in *Ss*-; p = 0.0091), CXCL1 (GM of 711.4 pg/ml in *Ss*+ vs. 1040 pg/ml in *Ss*-; p = 0.0318), CXCL2 (GM of 383.2 pg/ml in *Ss*+ vs. 578.1 pg/ml in *Ss*-; p = 0.0410), CXCL8 (GM of 291.1 pg/ml in *Ss*+ vs. 410.3 pg/ml in *Ss*-; p = 0.0468), CXCL9 (GM of 432.5 pg/ml in *Ss*+ vs. 620 pg/ml in *Ss*-; p = 0.0438), CXCL10 (GM of 347.7 pg/ml in *Ss*+ vs. 479.5 pg/ml in *Ss*-; p = 0.0430) and CXCL11 (GM of 313.2 pg/ml in *Ss*+ vs. 505 pg/ml in *Ss*-; p = 0.0360) in comparison with *Ss*- individuals. Thus, *Ss* infection also appears to modulate the pro-inflammatory chemokine milieu in T2DM.

## Anthelmintic treatment significantly increases the plasma cytokine levels in T2DM

To determine the effect of anthelmintic treatment on the pro-inflammatory cytokine milieu in T2DM, we measured the plasma levels of IL1α, IL-1β, IL-1Ra, IL-6, IL-12, IL-18, IL-23, IL-27, G-CSF and GM-CSF in *Ss*+ individuals before (pre-T) and 6 months after anthelmintic treatment (post-T). At post-T, the levels of IL1α (GM of 350.1 pg/ml in pre-T vs. 496 pg/ml in

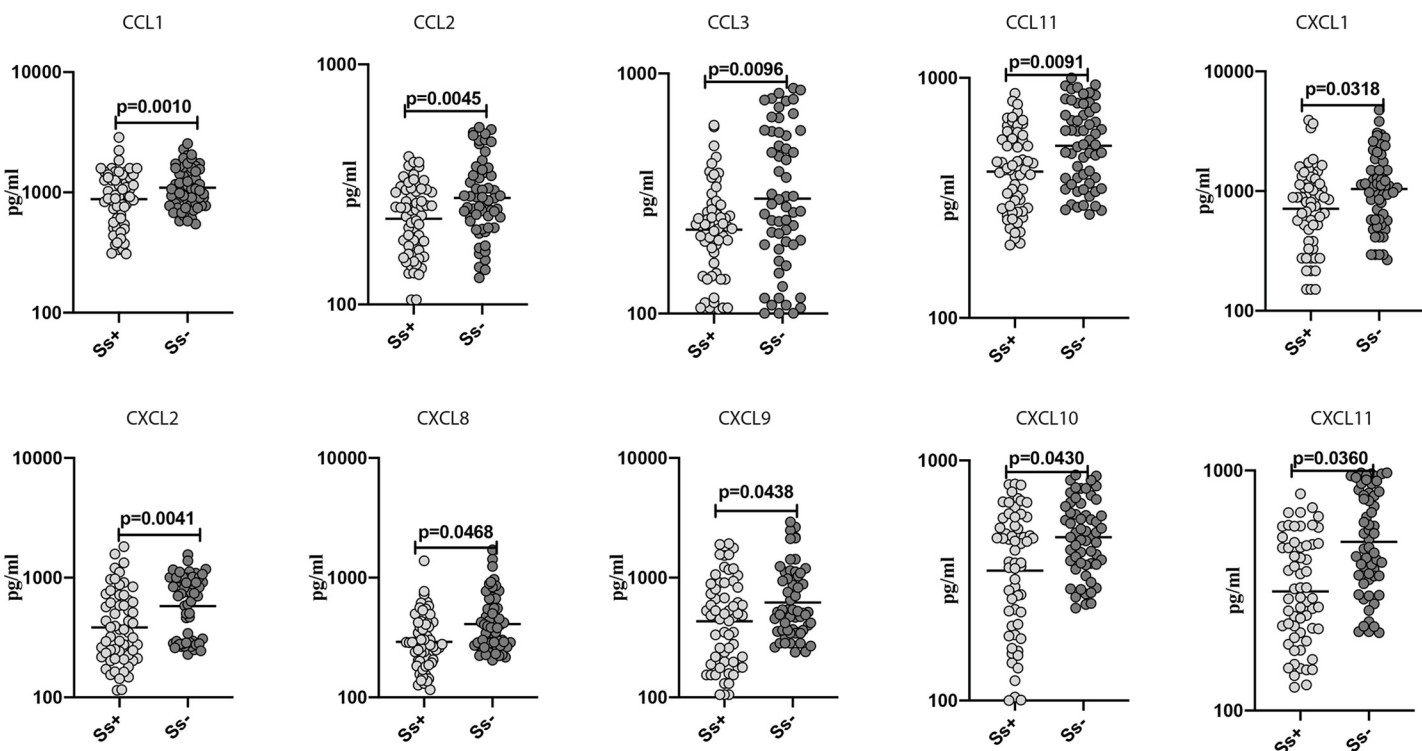

**Fig 2. Diminished plasma levels of pro-inflammatory chemokines in *Ss+* individuals with T2DM.** Plasma levels of CCL1, CCL2, CCL3, CCL11, CXCL1, CXL2, CXCL8, CXCL9, CXCL10 and CXCL11 chemokines were measured in *Ss+* and *Ss-* individuals. Each dot is an individual subject with the bar representing the geometric mean (GM). p values were calculated using the Mann-Whitney U tests followed by Holm's correction for multiple comparisons.

post-T; p = 0.0009), IL-1β (GM of 254.1 pg/ml in pre-T vs. 268.7 pg/ml in post-T; p = 0.0008), IL-6 (GM of 65.58 pg/ml in pre-T vs. 78.4 pg/ml in post-T; p = 0.0007), IL-12 (GM of 84.76 pg/ml in pre-T vs. 90.14 pg/ml in post-T; p = 0.0006), IL-18 (GM of 283.7 pg/ml in pre-T vs. 374.9 pg/ml in post-T; p = 0.0005), IL-23 (GM of 244.1 pg/ml in pre-T vs. 312.5 pg/ml in post-T; p = 0.0004), IL-27 (GM of 323.8 pg/ml in pre-T vs. 444.3 pg/ml in post-T; p = 0.0006), G-CSF (GM of 68.47 pg/ml in pre-T vs. 88.41 pg/ml in post-T; p = 0.0044) and GM-CSF (GM of 78.89 pg/ml in pre-Tvs. 92.99 pg/ml in post-T; p = 0.0427) were all significantly increased in comparison to pre-T levels (Fig 3). In contast, the levels of IL1Ra (GM of 267.6 pg/ml in pre-T vs. 250.2 pg/ml in post-T; p = 0.0001) was significantly decreased at post-T. Thus, anthelmintic treatment partially restores the pro-inflammatory cytokine milieu in T2DM.

## Anthelmintic treatment significantly increases the plasma chemokine levels in T2DM

To determine the effect of anthelmintic treatment on the pro-inflammatory chemokine milieu in T2DM, we measured the plasma levels of CCL1, CCL2, CCL3, CCL11, CXCL1, CXCL2, CXCL9, CXCL10 and CXCL11 in *Ss+* individuals before (pre-T) and 6 months after anthelmintic treatment (post-T). At post-T, the levels of CCL1 (GM of 878.3 pg/ml in pre-T vs. 939.2 pg/ml in post-T; p = 0.0009), CCL2 (GM of 227.4 pg/ml in pre-T vs. 272.1 pg/ml in post-T; p = 0.0008), CCL3 (GM of 223.6 pg/ml in pre-T vs. 282.7 pg/ml in post-T; p = 0.0007), CXCL1 (GM of 711.4 pg/ml in pre-T vs. 763.8 pg/ml in post-T; p = 0.0006), CXCL2 (GM of 383.2 pg/ml in pre-T vs. 512.9 pg/ml in post-T; p = 0.0005), CXCL8 (GM of 292.1 pg/ml in pre-T vs. 327.9 pg/ml in post-T; p = 0.0004), CXCL9 (GM of 432.5 pg/ml in pre-T vs. 546.6 pg/ml in

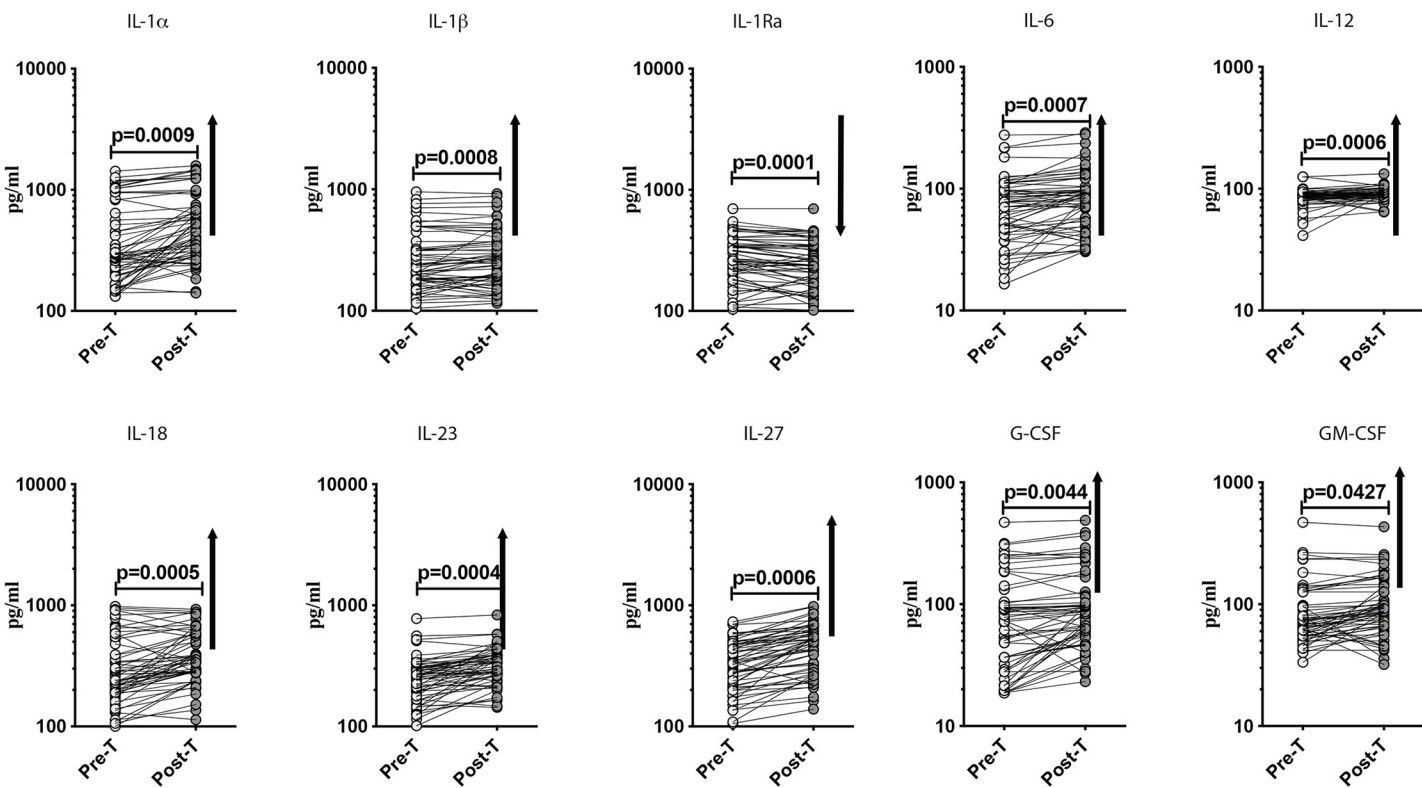

**Fig 3. Anthelmintic treatment significantly increased the plasma cytokine levels in T2DM.** Plasma levels of IL1-α, IL1-β, IL-1Ra, IL-6, IL-12, IL-18, IL-23, IL-27, G-CSF and GM-CSF cytokines were measured in *Ss+* individuals pretreatment [Pre-T] and 6 months following treatment [post-T] were measured. The data are represented as line graphs with each line representing a single individual. p values were calculated using the Wilcoxon signed rank test followed by Holm's correction for multiple comparisons.

post-T; p = 0.0009), CXCL10 (GM of 347.7 pg/ml in pre-T vs. 408.7 pg/ml in post-T; p = 0.0220) and CXCL11 (GM of 313.2 pg/ml in pre-T vs. 332.7 pg/ml in post-T; p = 0.0318) were all significantly increased in comparison to pre-T levels (Fig 4). Thus, anthelmintic treatment partially restores the pro-inflammatory chemokine milieu in T2DM.

## PCA analysis and heatmaps reveal trends in cytokine and chemokine milieu in helminth-T2DM co-morbidity

To assess the trends in cytokine and chemokine discrimination between *Ss+* (both pre- and post-treatment) and *Ss-* individuals, we plotted PCA with different inputs. As shown in Fig 5A, PCA analysis shows that cytokines and chemokines cluster differently between *Ss+* (pre-treatment) and *Ss-* individuals. In contrast, as shown in Fig 5B, PCA analysis shows very little clustering between *Ss+* (post-treatment) and *Ss-* individuals. Finally, heatmap analysis using the geometric mean values of cytokine and chemokine levels shows the highly upregulated expression profile in *Ss-* individuals and the moderately upregulated expression profile in post-T individuals compared to pre-T *Ss+* individuals (Fig 5C). Thus, these analysis help reveal the power of cytokines and chemokines to demarcate the effect of *Ss* infection on T2DM.

## Discussion

During the last few years, several studies have highlighted the important role played by the immune system in regulating metabolic homeostasis in both animal models and human

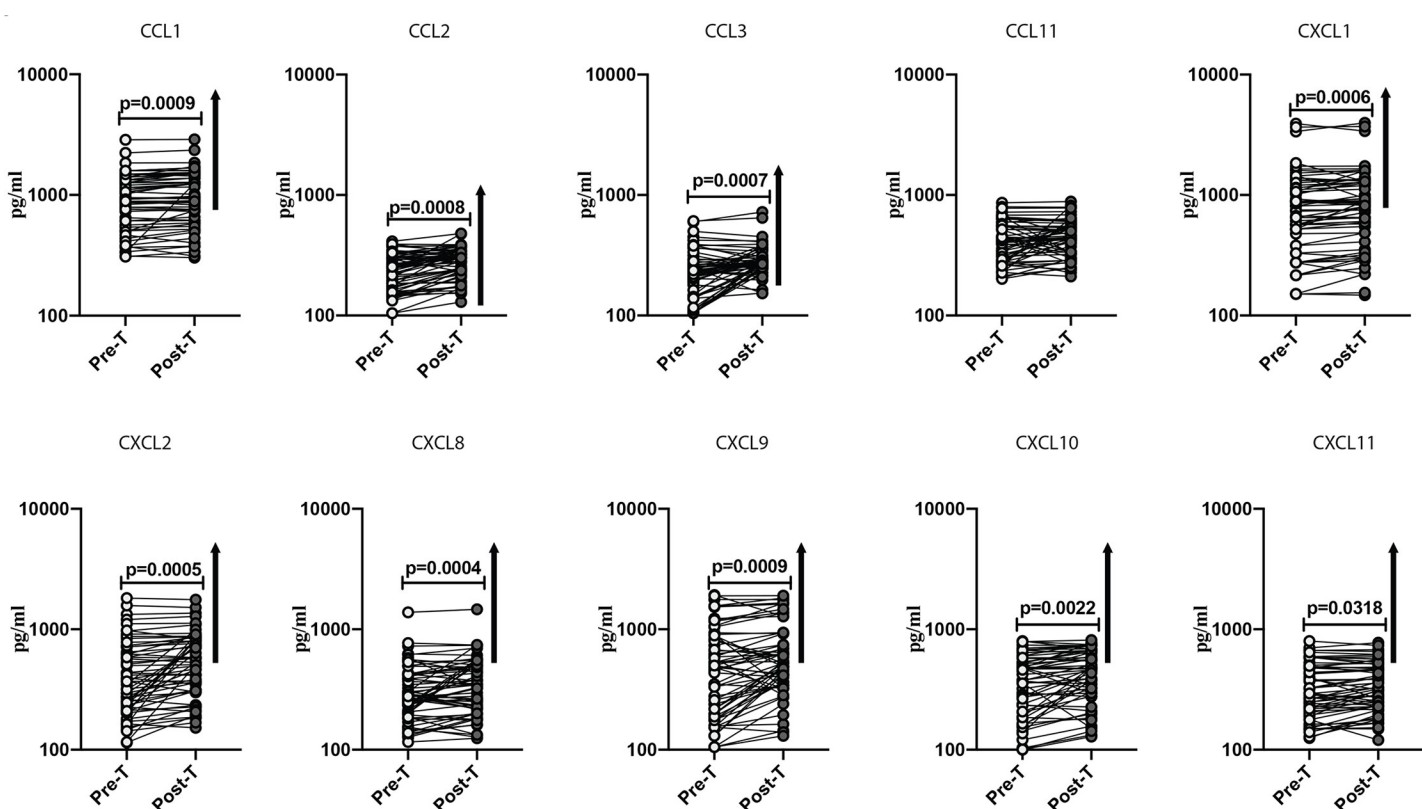

**Fig 4. Anthelmintic treatment significantly increased the plasma chemokine levels in T2DM.** Plasma levels of CCL1, CCL2, CCL3, CCL11, CXCL1, CXL2, CXCL8, CXCL9, CXCL10 and CXCL11 chemokines were measured in *Ss*+ individuals pretreatment [Pre-T] and 6 months following treatment [post-T] were measured. The data are represented as line graphs with each line representing a single individual. p values were calculated using the Wilcoxon signed rank test followed by Holm's correction for multiple comparisons.

disease [20, 21]. The cross-talk between innate and adaptive immune cell subsets on the one hand and the metabolic cells on the other leads to a complex network governing metabolic functions both at a systemic and tissue specific level [20, 21]. Interestingly, regulatory pathways induced by chronic helminth infections have been associated with reduced insulin resistance and a lower prevalence of metabolic syndrome and T2DM [4, 5, 7, 8, 13, 22], suggesting that helminth-mediated immunomodulation affords a protective effect against metabolic diseases [23].

Elevated levels of pro-inflammatory cytokines have been described in both cross-sectional and prospective studies in T2DM [24–26]. Moreover, heightened levels of IL-1β and IL-6 are predictive of T2DM [25, 27]. In addition, other pro-inflammatory cytokines such as IL-1α, IL-12 and IL-18 are linked to pancreatic islet inflammation [9, 28]. IL-23 is known to induce beta cell oxidative and ER stress in diabetic animal models [29], while IL-27 might play a pathogenic role [30], and IL-12 contributes to angiogenesis [31]. Similarly, GM-CSF and G-CSF also contribute to the pathogenesis of inflammation in T2DM [9, 28]. The elevated levels of these pro-inflammatory cytokines in T2DM is thought to reflect the activation of innate immune cells driven by increased nutrient concentrations. The sum total of these effects contributes to glucotoxicity, lipotoxicity, oxidative and ER stress of pancreatic islets in T2DM [10]. Similar to cytokines, a variety of chemokines are also involved in islet inflammation, oxidative and ER stress and dysglycemia. These include CC and CXC chemokines including CCL1, CCL2, CCL3, CCL11, CXCL1, CXCL2, CXCL9, CXCL10 and CXCL11 [9, 28]. Our data highlight a

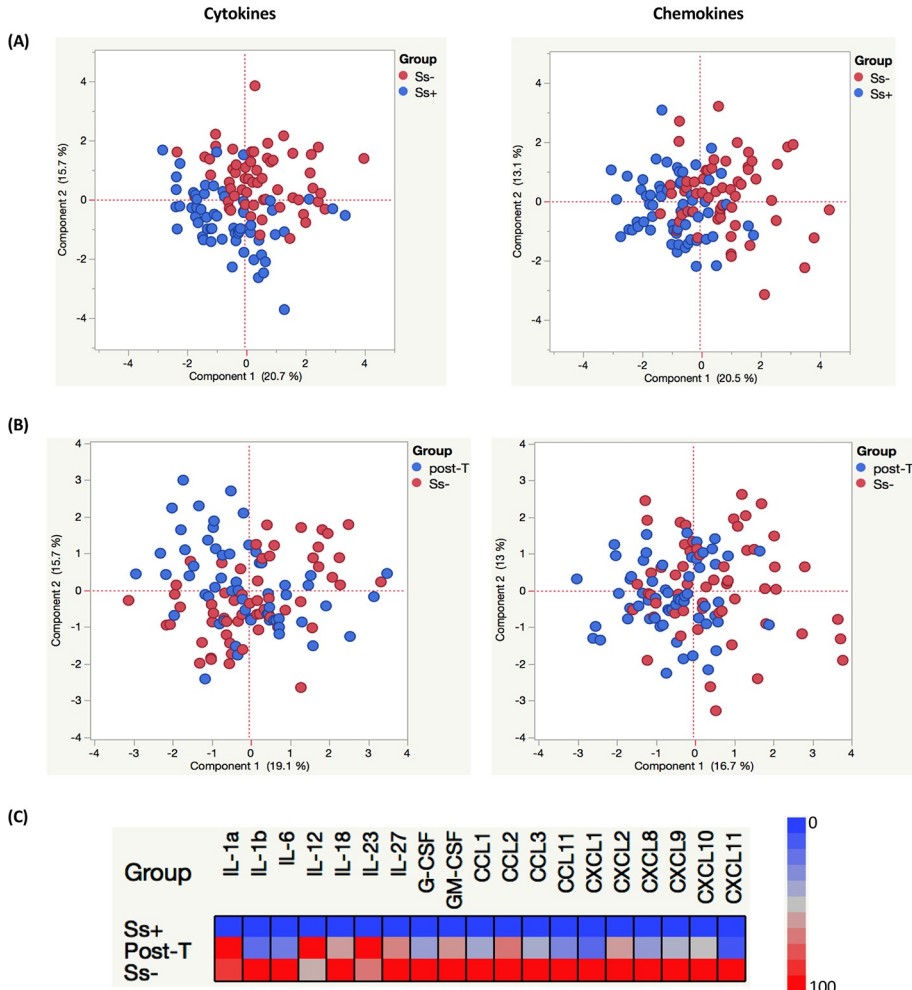

**Fig 5. Principal component Analysis (PCA) and heatmaps depicting circulating levels of cytokines and chemokines in *Ss*+ (pre- and post-treatment) and *Ss*- individuals. (A and B)** Principal component analysis (PCA) was performed to show the distribution of data from the combination of two groups *Ss*+ (blue circles) and *Ss*- (red circles)(A); and Post-T *Ss*+ (blue circles) and *Ss*- (red circles) (B). The PCA represents the two principal components of variation. (C) Heatmaps depicting the circulating levels of cytokines and chemokines in *Ss*+ (pre- and post-treatment) and *Ss*- individuals. Data (and scale) are log10 geometric mean fold change for each of the analytes measured for each of the groups. Plasma levels for *Ss*+, post treated and *Ss*- individuals were shown respectively, in which each row stands for a sample and each column stands for a cytokine and chemokines. Red color represents the highest values whereas blue color indicates the lowest values measured for each analyte.

novel feature of helminth–T2DM interaction in demonstrating the depression of circulating levels of all of the cytokines and chemokines mentioned above. Thus, coexistent chronic infection with *Ss* is associated with a dampened inflammatory cytokine and chemokine response in T2DM. Thus, our study provides a plausible biological mechanism for the observational studies showing a protective effect of helminth infection against T2DM.

While an inverse association has been reported between infection with *Schistosoma japonicum* and *Ss* and the prevalence of metabolic syndrome and T2DM, anthelmintic therapy was shown to impair metabolic homeostasis as characterized by increased homeostatic model assessment for insulin resistance (HOMA-IR) and haemoglobin A1c [4, 13, 22]. Our study again offers a plausible biological mechanism for this effect by demonstrating that anthelmintic treatment restores (at least partially) the elevated levels of pro-inflammatory cytokines and

chemokines in T2DM. As clearly demonstrated by the PCA and heatmap analysis, anthelmintic treatment alone does not cause a complete reversion to the systemic levels of cytokines and chemokines seen in helminth uninfected T2DM individuals. This is possibly due to the fact that helminth induced immunomodulation might require more time to return to homeostasis.

The limitations of the study include the absence of oral glucose tolerance testing for diabetes, the moderate sample size and the absence of testing for protozoa. Nevertheless, our study offers novel insights into the immunological interactions between helminth infections and metabolic disorders. In summary, our study demonstrates that *Ss* infection may provide a degree of protection from the pathology associated with T2DM by modulating the surrounding cytokine and chemokine milieu. Our data also suggest that helminth derived molecules or even helminth infection (per se) could offer novel therapeutic approaches to treating inflammatory metabolic diseases.

## Acknowledgments

We thank Dr. M. Satiswaran, B. Suganthi and Prabbu Balakrishnan for valuable assistance in collecting the clinical data for this study. We thank N. Pavan Kumar for technical assistance. We thank the staff of the Department of Epidemiology, NIRT, for valuable assistance in recruiting the patients for this study.

## Author Contributions

**Conceptualization:** Thomas B. Nutman, Subash Babu.

**Data curation:** Subash Babu.

**Formal analysis:** Anuradha Rajamanickam, Kannan Thiruvengadam, Subash Babu.

**Funding acquisition:** Thomas B. Nutman, Subash Babu.

**Investigation:** Anuradha Rajamanickam, Saravanan Munisankar.

**Methodology:** Anuradha Rajamanickam, Saravanan Munisankar, Chandrakumar Dolla, Kannan Thiruvengadam.

**Project administration:** Anuradha Rajamanickam, Saravanan Munisankar, Pradeep A. Menon.

**Resources:** Chandrakumar Dolla, Pradeep A. Menon.

**Supervision:** Thomas B. Nutman, Subash Babu.

**Validation:** Anuradha Rajamanickam.

**Writing – original draft:** Subash Babu.

**Writing – review & editing:** Thomas B. Nutman, Subash Babu.

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
