## [Decision Letter · Decision Letter 0]

19 Dec 2019

Dear Dr. Babu:

Thank you very much for submitting your manuscript "Helminth infection modulates systemic pro-inflammatory cytokines and chemokines implicated in Type 2 diabetes mellitus pathogenesis" (#PNTD-D-19-01779) for review by PLOS Neglected Tropical Diseases. Your manuscript was fully evaluated at the editorial level and by independent peer reviewers. The reviewers appreciated the attention to an important problem, but raised some substantial concerns about the manuscript as it currently stands. These issues must be addressed before we would be willing to consider a revised version of your study. We cannot, of course, promise publication at that time.

We therefore ask you to modify the manuscript according to the review recommendations before we can consider your manuscript for acceptance. Your revisions should address the specific points made by each reviewer. 

When you are ready to resubmit, please be prepared to upload the following:

(1) A letter containing a detailed list of your responses to the review comments and a description of the changes you have made in the manuscript.

(2) Two versions of the manuscript: one with either highlights or tracked changes denoting where the text has been changed (uploaded as a "Revised Article with Changes Highlighted" file); the other a clean version (uploaded as the article file).

(3) If available, a striking still image (a new image if one is available or an existing one from within your manuscript). If your manuscript is accepted for publication, this image may be featured on our website. Images should ideally be high resolution, eye-catching, single panel images; where one is available, please use 'add file' at the time of resubmission and select 'striking image' as the file type. 

Please provide a short caption, including credits, uploaded as a separate "Other" file. If your image is from someone other than yourself, please ensure that the artist has read and agreed to the terms and conditions of the Creative Commons Attribution License at http://journals.plos.org/plosntds/s/content-license (NOTE: we cannot publish copyrighted images). 

(4) If applicable, we encourage you to add a list of accession numbers/ID numbers for genes and proteins mentioned in the text (these should be listed as a paragraph at the end of the manuscript). You can supply accession numbers for any database, so long as the database is publicly accessible and stable. Examples include LocusLink and SwissProt.

(5) To enhance the reproducibility of your results, we recommend that you deposit your laboratory protocols in protocols.io, where a protocol can be assigned its own identifier (DOI) such that it can be cited independently in the future. For instructions see http://journals.plos.org/plosntds/s/submission-guidelines#loc-methods

While revising your submission, please upload your figure files to the Preflight Analysis and Conversion Engine (PACE) digital diagnostic tool, https://pacev2.apexcovantage.com/ PACE helps ensure that figures meet PLOS requirements. To use PACE, you must first register as a user. Then, login and navigate to the UPLOAD tab, where you will find detailed instructions on how to use the tool. If you encounter any issues or have any questions when using PACE, please email us at figures@plos.org.

We hope to receive your revised manuscript by Feb 17 2020 11:59PM. If you anticipate any delay in its return, we ask that you let us know the expected resubmission date by replying to this email.

To submit a revision, go to https://www.editorialmanager.com/pntd/ and log in as an Author. You will see a menu item call Submission Needing Revision. You will find your submission record there. 

Sincerely,

Mathieu Nacher

Guest Editor

Maria Periago

Deputy Editor

Reviewer's Responses to Questions

**Key Review Criteria Required for Acceptance?**

**Methods**

-Are the objectives of the study clearly articulated with a clear testable hypothesis stated?

-Is the study design appropriate to address the stated objectives?

-Is the population clearly described and appropriate for the hypothesis being tested?

-Is the sample size sufficient to ensure adequate power to address the hypothesis being tested?

-Were correct statistical analysis used to support conclusions?

-Are there concerns about ethical or regulatory requirements being met?

Reviewer #1: Do the author have data on CRP IL-1Ra? This inflammation markers are very robust and changes would support authors hypothesis

Reviewer #2: The work developed demonstrates that Ss+ individuals exhibited significantly diminished levels of the proinflammatory cytokines and chemokines. Anthelmintic treatment resulted in increased levels of all of the cytokines and chemokines, being a work of great relevance. The same research group published previous work with the same samples studied in: Clin Infect Dis. 2019 Aug 15; 69(4): 697–704. Metabolic Consequences of Concomitant Strongyloides stercoralis Infection in Patients With Type 2 Diabetes Mellitus with equal design and only differentiating in the studied cytokines. I suggest the author submit the article as short communication to show the other data found.

-Are the objectives of the study clearly articulated with a clear testable hypothesis stated? YES

-Is the study design appropriate to address the stated objectives? YES

- Is the population clearly described and appropriate for the hypothesis being tested? NO

- Is the sample size sufficient to ensure adequate power to address the hypothesis being tested? NO

- Were correct statistical analysis used to support conclusions? NO

- -Are there concerns about ethical or regulatory requirements being met? YES

- How was sample calculated? How were participants selected? 

- What parasitological method is used? How many samples were analyzed? What other methods were used to exclude other helminths? describe in more detail the methods used.

- The participants with strongyloides had no other helminths? Other studies verify the influence of other helminths and protozoa in dm2, being important to exclude other parasites or to include in the analyzes.

- Was only fasting glucose and HbA1c used to confirm diabetes? Why was the glucose tolerance test not performed? may have samples that are intolerant that they could have failed to include using fasting glucose alone.

- How dm1 and type 2 was differentiated? how can you be sure to be type 2? Clinical signs were considered to differentiate between dm1 and dm2.

- What is the rationale for hematological and other biochemical analyzes of the project? what purpose? This data was not used for anything.

- Justify the statistical analyzes studied at work. why was it used? why no logistic regression was performed to evaluate the effect of strongyloides on dm2

- For the analysis of the comparison of the groups in relation to the dosages, how was the comparison made? using average or median? looking at the figures some analysis seems that the confidence interval overlaps between the groups and would not have significant difference.

- Why was no analysis of clustered cytokines performed? What relationship between them?

Reviewer #3: -Are the objectives of the study clearly articulated with a clear testable hypothesis stated?

YES

-Is the study design appropriate to address the stated objectives?

YES

-Is the population clearly described and appropriate for the hypothesis being tested?

YES

-Is the sample size sufficient to ensure adequate power to address the hypothesis being tested?

YES

-Were correct statistical analysis used to support conclusions?

YES

-Are there concerns about ethical or regulatory requirements being met?

YES

**Results**

-Does the analysis presented match the analysis plan?

-Are the results clearly and completely presented?

-Are the figures (Tables, Images) of sufficient quality for clarity?

Reviewer #1: Figure 3 and 4 are visually not well presented. It would be helpful to see the mean values of pre and post, and may be have different colors for changes going up or down

Reviewer #2: -Does the analysis presented match the analysis plan? NO

- -Are the results clearly and completely presented? NO

- -Are the figures (Tables, Images) of sufficient quality for clarity? YES

Reviewer #3: -Does the analysis presented match the analysis plan?

YES

-Are the results clearly and completely presented?

YES

-Are the figures (Tables, Images) of sufficient quality for clarity?

YES

**Conclusions**

-Are the conclusions supported by the data presented?

-Are the limitations of analysis clearly described?

-Do the authors discuss how these data can be helpful to advance our understanding of the topic under study?

-Is public health relevance addressed?

Reviewer #1: Yes

Reviewer #2: -Are the conclusions supported by the data presented? NO

-Are the limitations of analysis clearly described? NO

-Do the authors discuss how these data can be helpful to advance our understanding of the topic under study? YES

-Is public health relevance addressed? NO

Reviewer #3: -Are the conclusions supported by the data presented?

YES, but the author did not measure any anti inflammatory markers

-Are the limitations of analysis clearly described?

NO, the author should add the limitations of thi study

-Do the authors discuss how these data can be helpful to advance our understanding of the topic under study?

YES

-Is public health relevance addressed?

YES, partially.

**Editorial and Data Presentation Modifications?**

Reviewer #1: See above my suggestions which I consider as minor

Reviewer #2: (No Response)

Reviewer #3: Method Section:

For the determination of T2DM status, the author used HbA1c and Random Blood Glucose. If possible, the author should also describe other factors that might affect HbA1c level in the study population eg anemia, etc.

For the measurement of cytokines level, the author should describe whether the measurements were done in paired samples ( SS pos and SS neg side by side for each sample)?

The author should also describe about other factors that might affect inflammatory markers, eg infections, medications, etc -- and whether this has been addressed by excluding those subjects or adjust for these factors in the analysis

**Summary and General Comments**

Reviewer #1: This is a very interesting observational study, which may be a starting point for further studies and drug development. The hypothesis is clear, the data are solid and the manuscript is well written. The major limitation is that the study is purely correlative, but it does not take away from its originality.

Reviewer #2: (No Response)

Reviewer #3: This is a well written article. The author describes the fact that helminth infections could modulate systemic pro-inflammatory cytokines and chemokines in newly diagnosed diabetes.

However, several issues need to be addressed.

Major

Why did the author did not measure any anti-inflammatory cytokines or chemokines? 

Minor comments

The author should also describe about other factors that might affect inflammatory markers, eg infections, medications, etc -- and whether this has been addressed by excluding those subjects or adjust for these factors in the analysis

For the measurement of cytokines level, the author should describe whether the measurements were done in paired samples ( SS pos and SS neg side by side for each sample)?

For the determination of T2DM status, the author used HbA1c and Random Blood Glucose. If possible, the author should also describe other factors that might affect HbA1c level in the study population eg anemia, etc.

PLOS authors have the option to publish the peer review history of their article (what does this mean?). If published, this will include your full peer review and any attached files.

Reviewer #1: Yes: Marc Donath

Reviewer #2: No

Reviewer #3: No

---

## [Editor Report · Decision Letter 1]

29 Jan 2020

Dear Dr. Babu,

We are pleased to inform you that your manuscript 'Helminth infection modulates systemic pro-inflammatory cytokines and chemokines implicated in Type 2 diabetes mellitus pathogenesis' has been provisionally accepted for publication in PLOS Neglected Tropical Diseases.

Before your manuscript can be formally accepted you will need to complete some formatting changes, which you will receive in a follow up email. A member of our team will be in touch within two working days with a set of requests.

Best regards,

Mathieu Nacher

Guest Editor

Maria Periago

Deputy Editor

---

## [Editor Report · Acceptance letter]

27 Feb 2020

Dear Dr. Babu,

We are delighted to inform you that your manuscript, "Helminth infection modulates systemic pro-inflammatory cytokines and chemokines implicated in Type 2 diabetes mellitus pathogenesis," has been formally accepted for publication in PLOS Neglected Tropical Diseases.

Best regards,

Serap Aksoy

Editor-in-Chief

Shaden Kamhawi

Editor-in-Chief
